# SLiMe: Segment Like Me

**Aliasghar Khani**[1,2], **Saeid Asgari Taghanaki**[1,2], **Aditya Sanghi**[1], **Ali Mahdavi Amiri**[2],
**Ghassan Hamarneh**[2]
[1] Autodesk Research
[2] School of Computing Science, Simon Fraser University

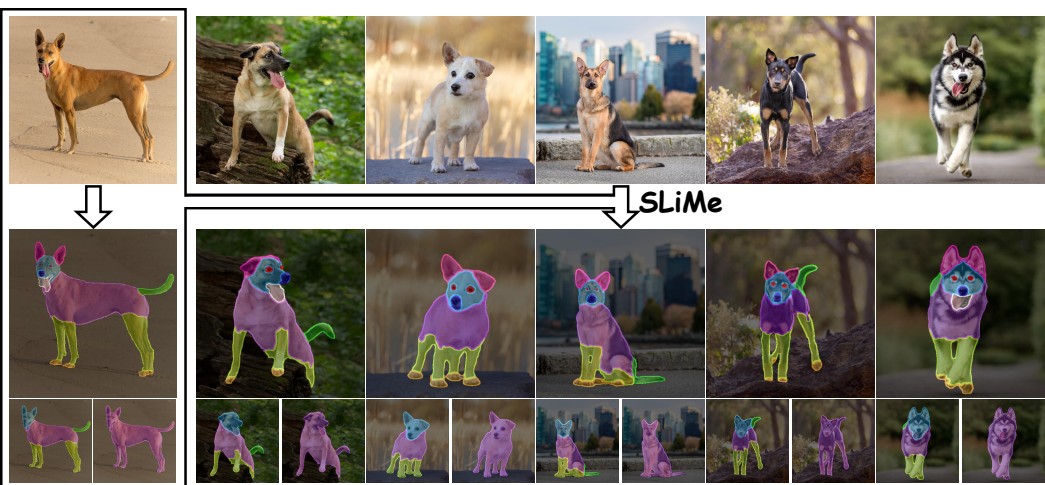

Figure 1: **SLiMe.** Using just one user-annotated image with various granularity (as shown in the leftmost column), *SLiMe* learns to segment different unseen images in accordance with the same granularity (as depicted in the other columns).

## Abstract

Significant advancements have been recently made using Stable Diffusion (SD), for a variety of downstream tasks, e.g., image generation and editing. This motivates us to investigate SD's capability for image segmentation at any desired granularity by using as few as only *one* annotated sample, which has remained largely an open challenge. In this paper, we propose *SLiMe*, a segmentation method, which frames this problem as a one-shot optimization task. Given a single image and its segmentation mask, we propose to first extract our novel *weighted accumulated self-attention map* along with cross-attention map from text-conditioned SD. Then, we optimize text embeddings to highlight areas in these attention maps corresponding to segmentation mask foregrounds. Once optimized, the text embeddings can be used to segment unseen images. Moreover, leveraging additional annotated data when available, i.e., few-shot, improves *SLiMe*'s performance. Through broad experiments, we examined various design factors and showed that *SLiMe* outperforms existing one- and few-shot segmentation methods. The source code of the project is publicly available.

## 1 Introduction

Image segmentation is a multifaceted problem, with solutions existing at various levels of granularity. For instance, in applications like expression recognition or facial alignment, segmenting images of faces into basic regions like nose and eyes might suffice. However, in visual effects applications, more detailed segments such as eye bags, forehead, and chin are necessary for tasks like wrinkle removal. Moreover, from the perspective of an end-user, a straightforward and effective approach to guide a segmentation method is determining what to segment and the desired level of detail across

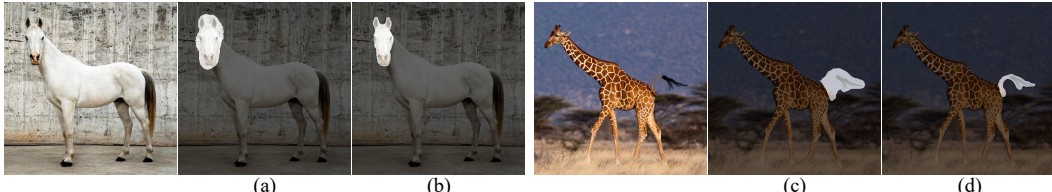

(a)     (b)        (c)    (d)

Figure 2: **Our proposed weighted accumulated self-attention maps' sample results.** Employing cross-attention naïvely without the self-attention for segmentation leads to inaccurate and noisy output (a and c). Using self-attention map along with cross-attention map to create WAS-attention map enhances the segmentation (b and d).

a broad set of images by providing only one or a few segmented examples for the method to use for training. Meanwhile, the user should not need to curate a large dataset with segmentation masks, train a large segmentation model, or encode elaborate and specific properties of target objects into the model. As a result, a customizable segmentation method that can adapt to different levels of granularity, using a few annotated samples, and provide users with the ability to intuitively define and refine the target segmentation according to their specific requirements, is of high importance.

Recent research has tackled the lack of segmentation data by delving into zero-shot, textual description based segmentation, and few-shot learning. DiffSeg (Tian et al., 2023) is a zero-shot segmentation method based on SD, which segments everything in the image. However, DiffSeg cannot be used to segment a specific object or part in the test images given a train sample, because its segmentation is not controllable in terms of which object to segment and segmentation granularity. Peekaboo (Burgert et al., 2022) is another work, which uses textual description for segmentation. To this end, given an image and a textual description of the target object to be segmented, they use Stable Diffusion and its loss function to optimize a randomly initialized segmentation mask to reach the desired mask. Nevertheless, it cannot be used to segmentation of test images given train images, because the textual description of the target object in each image is unique and is not transferrable. Another promising method is ReGAN (Tritrong et al., 2021). ReGAN first trains a GAN (Goodfellow et al., 2014) on the data of a specific class they aim to segment. Following this, they generate data by this GAN and the user manually annotates the generated data. Then both the generated data's features from the GAN and the annotations are utilized to train a segmentation model. In contrast, SegDDPM (Baranchuk et al., 2021) extracts features from a pre-trained diffusion model (DM) and trains an ensemble of MLPs for segmentation using few labeled data. Both excel in segmentation with 10-50 examples but struggle with extremely limited samples. Furthermore, these models require training on data specific to each category. For instance, to segment horses, it is necessary to collect a large dataset of horse images, a task that can be inherently cumbersome.

Whereas, SegGPT (Wang et al., 2023) employs one-shot learning, training on color-randomized segmentation data which includes both instance and part-level masks. During inference, it segments only one region in a target image using a reference image and its binary segmentation mask. While SegGPT is effective, it demands a significant amount of annotated segmentation data for initial training, keeping the challenge of training effectively with a single annotation still unaddressed.

In this paper, we propose Segment Like Me (*SLiMe*), which segments any object/part from the same category based on a given image and its segmentation mask with an arbitrary granularity level in a one-shot manner, avoiding the need for extensive annotated segmentation data or training a generative model like GAN for a specific class (see Figure 1 and Figure 8 for some examples). For this purpose, we leverage the rich knowledge of existing large-scale pre-trained vision/language model, Stable Diffusion (SD) (Rombach et al., 2022a). Recent studies like (Hertz et al., 2022) have shown that the cross-attention maps of models like SD highlight different regions of the image when the corresponding text changes. This property has been utilized to modify generated images (Hertz et al., 2022) and to achieve image correspondence (Hedlin et al., 2023). Expanding on this idea, we present two key insights. First, the multifaceted segmentation problem can be framed as a one-shot optimization task where we fine-tune the text embeddings of SD to capture semantic details such as segmented regions guided by a reference image and its segmentation mask, where each text embedding corresponds to a distinct segmented region. Second, we observed that using standalone cross-attention maps lead to imprecise segmentations, as depicted in Figure 2. To rectify this, we propose a novel weighted accumulated self (WAS)-attention map (see Section 4). This

attention map incorporates crucial semantic boundary information and employs higher-resolution self-attention maps, ensuring enhanced segmentation accuracy.

Based on these insights, *SLiMe* uses a single image and its segmentation mask to fine-tune SD's text embeddings through cross- and WAS-attention maps. These refined embeddings emphasize segmented regions within these attention maps, and are used to segment real-world images during inference, mirroring the granularity of the segmented region from the image used for optimization. Through various quantitative and qualitative experiments, we highlight the efficacy of our approach. *SLiMe*, even when reliant on just one or a handful of examples, proves to be better or comparable to supervised counterparts demanding extensive training. Furthermore, despite not being trained on a specific category, *SLiMe* outperforms other few-shot techniques on average and on most parts, across almost all the datasets. For instance, we outperform ReGAN (Tritrong et al., 2021) by nearly 10% and SegDDPM (Baranchuk et al., 2021) by approximately 2% in a 10-sample setting. Additionally, in a 1-sample context, we exceed SegGPT by around 12% and SegDDPM by nearly 11%.

## 2    RELATED WORK

**Semantic Part Segmentation.** In computer vision, semantic segmentation, wherein a class label is assigned to each pixel in an image, is an important task with several applications such as scene parsing, autonomous systems, medical imaging, image editing, environmental monitoring, and video analysis (Sohail et al., 2022; He et al., 2016; Chen et al., 2017a; Zhao et al., 2017; He et al., 2017; Chen et al., 2017b; Sandler et al., 2018; Chen et al., 2018). A more fine-grained derivative of semantic segmentation is semantic part segmentation, which endeavors to delineate individual components of objects rather than segmenting the entirety of the objects. Algorithms tailored for semantic part segmentation find applications in subsequent tasks such as pose estimation (Zhuang et al., 2021), activity analysis (Wang & Yuille, 2015), object re-identification (Cheng et al., 2016), autonomous driving and robot navigation (Li et al., 2023). Despite notable advancements in this domain (Li et al., 2023; 2022), a predominant challenge faced by these studies remains the substantial need for annotated data, a resource that is often difficult to procure. Hence, to address these challenges, research has pivoted towards exploring alternative inductive biases and supervision forms. However, a limitation of such methodologies is their reliance on manually curated information specific to the object whose parts they aim to segment. For example, authors of (Wang & Yuille, 2015) integrate inductive biases by harnessing edge, appearance, and semantic part cues for enhanced part segmentation. Compared to these approaches, our method only necessitates a single segmentation mask and doesn't rely on ad-hoc inductive biases, instead leveraging the knowledge embedded in SD.

**Few-shot Semantic Part Segmentation.** One approach to reduce the need for annotated data is to frame the problem within the few-shot part segmentation framework. There is a large body of work on few-shot semantic segmentation (Catalano & Matteucci, 2023; Xiong et al., 2022; Johnander et al., 2022; Zhang et al., 2022; Li et al., 2022), however, they mostly focus on the object- (not part-) level. A recent paper, ReGAN (Tritrong et al., 2021), proposed a few-shot method for part segmentation. To achieve this, the researchers leveraged a large pre-trained GAN, extracting features from it and subsequently training a segmentation model using these features and their associated annotations. While this approach enables the creation of a semantic part segmentation model with limited annotated data, it suffers from a drawback. Specifically, to train a model to segment parts of a particular object category, first a GAN is required to be trained from scratch on data from the same category. For instance, segmenting parts of a human face would necessitate a GAN trained on generating human face images. Thus, even though the method requires minimal annotated data, it demands a substantial amount of images from the relevant category. Following that, a few images, which are generated by the GAN, need to be manually annotated to be used for training the segmentation model. Afterward, a multitude of images should be generated by the GAN and segmented by the trained segmentation model. Finally, all the annotated data and pseudo-segmented data are used for training a segmentation model from scratch. Instead, we leverage pre-trained DMs that are trained on large general datasets, eliminating the need to curate category-specific datasets.

**Diffusion models for semantic part segmentation.** DMs (Sohl-Dickstein et al., 2015) are a class of generative models that have recently gained significant attention because of their ability to generate high-quality samples. DMs have been used for discriminative tasks such as segmentation, as shown

in SegDDPM (Baranchuk et al., 2021). Given a few annotated images, they use internal features of DM, to train several MLP modules, for semantic part segmentation. Compared to SegDDPM, we utilize the semantic knowledge of text-conditioned SD, and just optimize the text embeddings. This way, we have to optimize fewer parameters for the segmentation task, which makes it possible to optimize using just one segmentation sample.

SD (Rombach et al., 2022a) has been used for several downstream tasks such as generating faithful images (Chefer et al., 2023), inpainting, outpainting (Rombach et al., 2022a), generating 3D shapes using text (Tang, 2022), and editing images guided by a text prompt (Brooks et al., 2023). In addition to these, a large body of work fine-tune SD or use its cross-attention modules to perform interesting tasks. For instance, (Gal et al., 2022) fine-tunes SD's text embeddings to add a new object or style to its image generation space. Another example, (Hertz et al., 2022) uses SD's cross-attention modules to impose more control over the generation process. Moreover, in a third instance, authors of (Mokady et al., 2023) edit a real image using SD's cross-attention modules. SD's cross-attention maps have been used for image correspondence by (Hedlin et al., 2023). Lastly, a recent paper (Patashnik et al., 2023), uses SD's self-attention and cross-attention modules for object level shape variations. Although these papers explore the applicability of SD in different tasks, its utilization in semantic part segmentation is not fully explored. Therefor, in this work, we take advantage of SD's self-attention and cross-attention modules and fine-tune its text embeddings through these attention mechanisms to perform semantic part segmentation even with just one annotated image.

## 3 BACKGROUND

**Latent Diffusion Model (LDM)**. One category of generative models are LDMs, which model the data distribution by efficiently compressing it into the latent space of an autoencoder and utilizing a DM to model this latent space. An appealing feature of LDMs is that their DM, denoted as $\epsilon(.; \theta)$, can be extended to represent conditional distributions, conditioned on text or category. To train a text-conditioned LDM, a natural language prompt is tokenized to obtain $P$. Then $P$ is passed to a text encoder $\mathcal{G}(.; \theta)$ to get $\mathcal{P} = \mathcal{G}(P; \theta)$. Alternatively, it is possible to obtain $\mathcal{P}$ by randomly initializing a tensor of the same size. Afterward, the input image $I$ is encoded to obtain $\mathcal{I}$, and a standard Gaussian noise $\epsilon$ is added to it with respect to time step $t$ to get $\mathcal{I}_t$. Finally, the following objective is used to optimize the parameters of both $\mathcal{G}(.; \theta)$ and $\epsilon(.; \theta)$, with the aim of enabling the model to acquire the capability to predict the added noise $\epsilon$:

$$\mathcal{L}_{LDM} = \mathbb{E}_{\mathcal{I}, \epsilon \sim \mathcal{N}(0,1), t}[\|\epsilon - \epsilon(\mathcal{I}_t, t, \mathcal{P}; \theta)\|_2^2]. \tag{1}$$

In this work, we use text-conditioned SD (Rombach et al., 2022b), as our LDM, for two reasons. First, SD is conditioned on the text using the cross-attention modules, which have shown to exhibit rich semantic connections between the text and the image embeddings (Hertz et al., 2022). Second, the internal features of SD are semantically meaningful and preserve the visual structure of the input image, enhancing the interrelation between text and image.

**Attention Modules**. SD's DM employs a UNet structure, which has two types of attention modules (Vaswani et al., 2017): self-attention and cross-attention. The self-attention module calculates attention across the image embedding, capturing relationships between a specific element and other elements within the same image embedding. On the other hand, the cross-attention module computes relationships between the latent representations of two different modalities, like text and image in the case of text-conditioned SD.

An attention module comprises three components: query, key, and value. It aims to transform the query into an output using the key-value pair. Therefore, given query $Q$, key $K$, and value $V$ vectors with the dimension of $d$, the output $O$ of an attention module is defined as follows:

$$O = \text{Softmax}\left(\frac{QK^\intercal}{\sqrt{d}}\right) \cdot V. \tag{2}$$

In the self-attention module, the query, key, and value vectors are derived from the image embedding, while in the cross-attention module, the query vector is derived from the image embedding, and

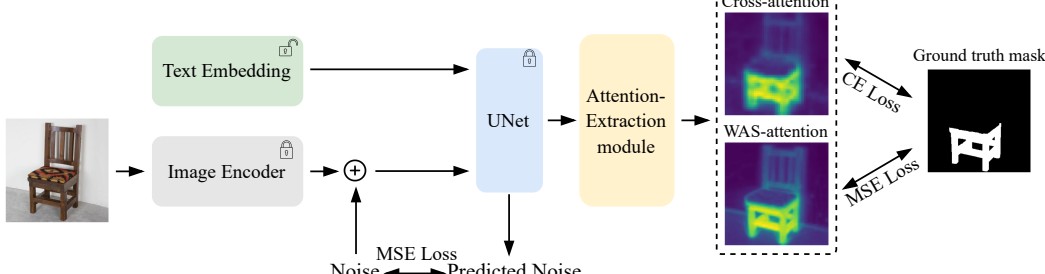

Figure 3: **Optimization step.** After extracting image embeddings and adding noise, we pass them, along with a text embedding obtained either by using a text encoder or initialized randomly, through the UNet to obtain cross- and WAS-attention maps. Two losses are then calculated using these maps and the ground truth mask. Additionally, SD's loss is incorporated from comparing the added noise with the UNet's predicted noise.

the key and value vectors are derived from the text embedding. In our scenario, we extract the normalized attention map denoted as $S = \text{Softmax}\left(\frac{QK^{\intercal}}{\sqrt{d}}\right)$, which is applicable to both the self-attention and cross-attention modules, and we note them as $S_{sa} \in \mathbb{R}^{H' \times W' \times H' \times W'}$ and $S_{ca} \in \mathbb{R}^{H' \times W' \times T}$, respectively. In this context, $H'$ and $W'$ represent the height and width of the image embedding and $T$ denotes the total number of text tokens. $S_{sa}$ shows the pairwise similarity of the elements in its input image embedding. Hence, each element $p$ in its input, is associated with an activation map, highlighting the similar elements to $p$ (Patashnik et al., 2023). Moreover, the intensity of the similar elements decrease as we move farther away from $p$. On the other hand, for each text token, $S_{ca}$ has an activation map, which effectively spotlights elements within the image embedding that align with that token within the model's semantic space. For example, if the model is instructed to generate an image of a bear with the text prompt "a bear", the activation map associated with "bear" token within $S_{ca}$, will emphasize on those elements that correspond to the bear object within the generated image.

## 4 METHOD

We introduce *SLiMe*, a method that enables us to perform segmentation at various levels of granularity, needing only one image and its segmentation mask. Prior research has demonstrated that SD's cross-attention maps can be used in detecting coarse semantic objects during the generation process for more control in generation (Hertz et al., 2022) or finding correspondence between images (Hedlin et al., 2023). However, there remains uncertainty regarding the applicability of cross-attention maps for finer-grained segmentation of objects or parts, especially within real-world images. To resolve this, we frame the segmentation problem as a one-shot optimization task where we extract the cross-attention map and our novel WAS-attention map to fine-tune the text embeddings, enabling each text embedding to grasp semantic information from individual segmented regions (Figure 3). During the inference phase, we use these optimized embeddings to obtain the segmentation mask for unseen images. In what follows, we will first delve into the details of the text embedding optimization and then the inference process.

### 4.1 OPTIMIZING TEXT EMBEDDING

Given a pair of an image ($I \in \mathbb{R}^{H \times W \times 3}$) and a segmentation mask ($M \in \{0, 1, 2, ..., K-1\}^{H \times W}$) with $K$ classes, we optimize the text embeddings using three loss terms. The first loss term is a cross entropy loss between the cross-attention map and the ground truth mask. The second one, is the Mean Squared Error (MSE) loss between the WAS-attention map and the ground truth mask. These loss terms refine the text embeddings and enable them to learn to emphasize segmented regions within both cross- and WAS-attention maps. Additionally, there is a subsequent SD regularization term to ensure that the optimized text embeddings remain within the trained distribution of SD.

To optimize the text embeddings, it is necessary to extract the cross-attention and self-attention maps. These maps are derived from SD's UNet by initially encoding the training image $I$ into the image embedding, $\mathcal{I}$. Subsequently, a standard Gaussian noise is added to this embedding with respect to the time step $t_{\text{opt}}$, resulting in $\mathcal{I}_t$. Next, a text prompt is converted to a sequence of

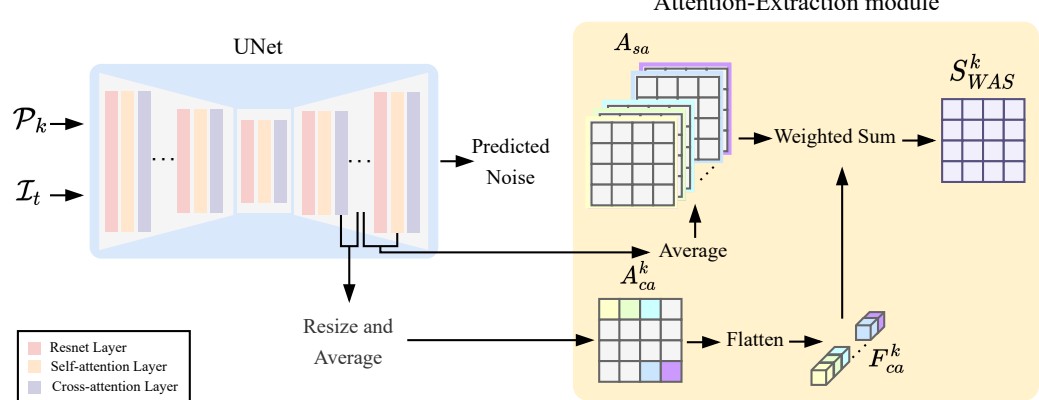

Figure 4: **Attention-Extraction module.** To extract WAS-attention map of $k^{th}$ text embedding with respect to an image, we follow these three steps: (1) We feed the $k^{th}$ text embedding ($\mathcal{P}_k$) together with the noised embedding of the image ($\mathcal{I}_t$) to the UNet. Then calculate $A_{ca}^k$ by extracting the cross-attention maps of $\mathcal{P}_k$ from several layers, resizing and averaging them. (2) We extract the self-attention maps from several layers and average them ($A_{sa}$). (3) Finally, we flatten $A_{ca}^k$ to get $F_{ca}^k$ and calculate a weighted sum of channels of $A_{sa}$, by weights coming from $F_{ca}^k$, and call it "Weighted Accumulated Self-attention map" ($S_{WAS}^k$). The UNet also produces an output that represents the predicted noise, which is used for calculating the loss of the SD.

text embeddings denoted as $\mathcal{P}$. We then take the first $K$ text embeddings and optimize them. The corresponding text embedding of each class is denoted by $\mathcal{P}_k$. It is essential to note that SD is configured to handle 77 text tokens. Consequently, our method can accommodate up to 77 segmentation classes, which is sufficient for most applications. Finally, $\mathcal{P}$ and $\mathcal{I}_t$ are fed into the UNet to obtain the denoised image embedding $\mathcal{I}'$ and extract the cross- and self-attention maps.

SD has multiple cross-attention modules distributed across various layers. We denote the normalized cross-attention map of the $l^{th}$ layer as $\{S_{ca}\}_l \in \mathbb{R}^{H'_l \times W'_l \times T}$ and average them over different layers, as we have empirically observed that this averaging improves the results. However, since $H'_l$ and $W'_l$ vary across different layers, we resize all $\{S_{ca}\}_l$ to a consistent size for all the utilized layers. Finally, the attention map employed in our loss function is calculated as follows:

$$A_{ca} = Average_l(Resize(\{S_{ca}\}_l)), \tag{3}$$

where $A_{ca} \in \mathbb{R}^{H'' \times W'' \times T}$, $Average_l$ computes the average across layers, and $Resize$ refers to bilinear interpolation for resizing to dimensions $H'' \times W''$. Figure 4 visually depicts this procedure. Finally, we compute the cross-entropy loss between the resized ground truth mask $M$ to $H'' \times W''$ (referred to as $M'$) and first $K$ channels in the resized cross-attention map $A_{ca}$ for $k = \{0, ..., K - 1\}$, as outlined below:

$$\mathcal{L}_{CE} = CE(A_{ca}^{[0:K-1]}, M'), \tag{4}$$

where *CE* refers to cross-entropy. Using this loss, we optimize $k^{th}$ text embedding such that $A_{ca}^k$ highlights the $k^{th}$ class's region in the segmentation mask, for $k = \{1, ..., K - 1\}$. Note that we do not optimize the first text embedding and assign $A_{ca}^0$ to the background class, as empirically we have found that optimizing it yields suboptimal performance.

However, as the resolution of $\{S_{ca}\}_l$ we use are lower than the input image, object edges are vague in them. To enhance segmentation quality, we propose WAS-attention map, which integrates both self-attention and cross-attention maps. Besides possessing pairwise similarity between the image embedding's elements, the self-attention map has two additional features that make it suitable to be used for improving the segmentation results. First, the self-attention maps that we use, have higher resolution of feature maps compared to utilized cross-attention maps. Second, it shows the boundaries in more detail. Table 1, shows the importance of using the WAS-attention map which yields an average improvement of $6.0\%$ in terms of mIoU over simply using the cross-attention map for generating the segmentation mask. Like the cross-attention maps, we extract self-attention maps from multiple layers and compute their average as follows:

$$A_{sa} = Average_l(\{S_{sa}\}_l), \tag{5}$$

where $A_{sa} \in \mathbb{R}^{H'_l \times W'_l \times H'_l \times W'_l}$ and $Average_l$ calculates the average across layers. In equation 5 there is no need for a $Resize$ function as the self-attention maps that we use, all have the same size.

Table 1: **Ablating the effect of WAS-attention.** These numerical results, underscore the crucial contribution of WAS-attention maps to the quality of *SLiMe*'s outcomes.

| Use WAS-Attention Map | Body | Light | Plate | Wheel | Window | BG | Average |
|---|---|---|---|---|---|---|---|
| ✗ | $77.8 \pm 0.2$ | $48.2 \pm 2.5$ | $44.1 \pm 4.2$ | $63.9 \pm 0.1$ | $66.9 \pm 0.2$ | $75.3 \pm 0.2$ | $62.7 \pm 1.3$ |
| ✓ | $\mathbf{81.5 \pm 1.0}$ | $\mathbf{56.8 \pm 1.2}$ | $\mathbf{54.8 \pm 2.7}$ | $\mathbf{68.3 \pm 0.1}$ | $\mathbf{70.3 \pm 0.9}$ | $\mathbf{78.4 \pm 1.6}$ | $\mathbf{68.3 \pm 1.0}$ |

To calculate WAS-attention map, we first resize $A_{ca}^k$ to match the size of $A_{sa}$ using bilinear interpolation and call it $R_{ca}^k$. Consequently, for each element $p$ in $R_{ca}^k$ we have a channel in $A_{sa}$ that highlights relevant elements to $p$. Finally, we calculate the weighted sum of channels of $A_{sa}$ to obtain $S_{WAS}^k$ (WAS-attention map). The weight assigned to each channel is the value of the corresponding element of that channel in $R_{ca}^k$ (Figure 4). This process can be outlined as follows:

$$S_{WAS}^k = sum(flatten(R_{ca}^k) \odot A_{sa}). \tag{6}$$

This refinement enhances the boundaries because $A_{sa}$ possesses rich understanding of the semantic region boundaries (see the cross-attention and WAS-attention maps in Figure 3). At the end, we resize $S_{WAS}^k$ to $H'' \times W''$ and calculate the *MSE* loss this way:

$$\mathcal{L}_{MSE} = \sum_{k=0}^{K-1} \|Resize(S_{WAS}^k) - M_k'\|_2^2, \tag{7}$$

where $M_k'$ is a binary mask coming from the resized ground truth mask $M'$, in which only the pixels of the $k^{th}$ class are 1.

The last loss we use is the SD's loss function ($\mathcal{L}_{LDM}$), which is the *MSE* loss between the added noise and the predicted noise. We use this loss to prevent the text embeddings from going too far from the understandable space by SD. Finally, our objective to optimize the text embeddings is defined as:

$$\mathcal{L} = \mathcal{L}_{CE} + \alpha\mathcal{L}_{MSE} + \beta\mathcal{L}_{LDM}, \tag{8}$$

where $\alpha$ and $\beta$ are the coefficients of the loss functions.

## 4.2 INFERENCE

During inference, our objective is to segment unseen images at the same level of details as the image used during optimization. To achieve this, we begin with the unseen image and encode it into the latent space of SD. Following this, a standard Gaussian noise is introduced to the encoded image, with the magnitude determined by the time parameter $t_{\text{test}}$. Subsequently, we use the optimized text embeddings along with the encoded image to derive corresponding cross-attention and self-attention maps from the UNet model. These attention maps, as shown in Figure 4, enable us to obtain WAS-attention maps for each text embedding. Afterward, we select the first $K$ WAS-attention maps that correspond to $K$ classes. These selected maps are then resized using bilinear interpolation to match the dimensions of the input image and are stacked along the channel dimension. Subsequently, we generate a segmentation mask by performing an argmax across the channels. It is important to note that this process can be repeated for multiple unseen images during inference, without requiring a new optimization. An analysis of the selection of various parameters used in our method is provided in the Appendix A.2.

## 5 EXPERIMENTS

In this section, we demonstrate the superiority of *SLiMe* in semantic part segmentation. We use mIoU to compare our approach against three existing methods: ReGAN (Tritrong et al., 2021), SegDDPM (Baranchuk et al., 2021), and SegGPT (Wang et al., 2023) on two datasets: PASCAL-Part (Chen et al., 2014) and CelebAMask-HQ (Lee et al., 2020). ReGAN and SegDDPM utilize pre-trained GAN and DDPM models, respectively, training them on FFHQ and LSUN-Horse datasets for face and horse part segmentation. Additionally, ReGAN employs a pre-trained GAN from the LSUN-Car dataset for car part segmentation. We present the results for both 10-sample and 1-sample

Table 2: **Segmentation results for class car.** *SLiMe* consistently outperforms ReGAN, even though ReGAN utilized generated data alongside 10 annotated data for training. Furthermore, our method exhibits superior performance to SegGPT on average, despite SegGPT being supervised. The first two rows show the supervised methods, for which we use the reported numbers in ReGAN. The second two rows show the 10-sample setting and the last two rows, refer to the 1-sample scenario. ⋆ indicates the supervised methods.

| | Body | Light | Plate | Wheel | Window | Background | Average |
|---|---|---|---|---|---|---|---|
| CNN⋆ | 73.4 | 42.2 | 41.7 | 66.3 | 61.0 | 67.4 | 58.7 |
| CNN+CRF⋆ | 75.4 | 36.1 | 35.8 | 64.3 | 61.8 | 68.7 | 57 |
| ReGAN | 75.5 | 29.3 | 17.8 | 57.2 | 62.4 | 70.7 | 52.15 |
| *SLiMe* | **81.5 ± 1.0** | **56.8 ± 1.2** | **54.8 ± 2.7** | **68.3 ± 0.1** | **70.3 ± 0.9** | **78.4 ± 1.6** | **68.3 ± 1.0** |
| SegGPT⋆ | 62.7 | 18.5 | 25.8 | **65.8** | **69.5** | **77.7** | 53.3 |
| *SLiMe* | **79.6 ± 0.4** | **37.5 ± 5.4** | **46.5 ± 2.6** | 65.0 ± 1.4 | 65.6 ± 1.6 | 75.7 ± 3.1 | **61.6 ± 0.5** |

Table 3: **Segmentation results for class horse.** *SLiMe* outperforms ReGAN, SegDDPM, and SegGPT on average and most of the parts. The first two rows show the supervised methods, for which we use the reported numbers in ReGAN. The middle three rows show the 10-sample setting and the last three rows, are the results of the 1-sample scenario. ⋆ indicates the supervised methods.

| | Head | Leg | Neck+Torso | Tail | Background | Average |
|---|---|---|---|---|---|---|
| Shape+Appereance⋆ | 47.2 | 38.2 | 66.7 | - | - | - |
| CNN+CRF⋆ | 55.0 | 46.8 | - | 37.2 | 76 | - |
| ReGAN | 50.1 | 49.6 | **70.5** | 19.9 | 81.6 | 54.3 |
| SegDDPM | 41.0 | 59.1 | 69.9 | 39.3 | **84.3** | 58.7 |
| *SLiMe* | **63.8 ± 0.7** | **59.5 ± 2.1** | 68.1 ± 4.4 | **45.4 ± 2.4** | 79.6 ± 2.5 | **63.3 ± 2.4** |
| SegGPT⋆ | 41.1 | 49.8 | **58.6** | 15.5 | 36.4 | 40.3 |
| SegDDPM | 12.1 | 42.4 | 54.5 | 32.0 | 74.1 | 43.0 |
| *SLiMe* | **61.5 ± 1.0** | **50.3 ± 0.7** | 55.7 ± 1.1 | **40.1 ± 2.9** | **74.4 ± 0.6** | **56.4 ± 0.8** |

settings, utilizing a single validation sample for 10-sample experiments of *SLiMe*. Also, all experiments are conducted three times with different initializations, reporting their mean and standard deviation. We conduct experiments for SegDDPM and SegGPT using the custom version of test sets of the above-mentioned datasets, which are based on ReGAN settings, and report their results accordingly. For the remaining methods, we reference the results reported by ReGAN. Note that ReGAN and SegDDPM are not universally applicable to arbitrary classes, unless a large dataset for the given class is collected and a generative model is trained. However, *SLiMe* does not require collecting large category specific data and training an additional generative model, because of the inherent semantic knowledge embedded in SD (Figure 8). Whereas SegGPT requires a large segmentation dataset to be trained initially.

**PASCAL-Part**. This dataset provides detailed annotations of object parts. For our experiments, we focus on car and horse classes (for more details, please refer to Appendix B.1). Table 2 presents results for the car class. As there is no available pre-trained model for the car class in SegDDPM, we couldn't make a comparison with this model for this category. As evident from Table 2, *SLiMe* outperforms ReGAN in the 10-sample setting on average and all the part segments by a significant margin. Moreover, in the 1-sample setting, *SLiMe* either outperforms SegGPT by a large margin or performs comparably. Likewise, Table 3 displays our results for the horse class, where it is evident that our method, *SLiMe*, outperforms ReGAN, SegDDPM, and SegGPT on average and for most of the parts. It is worth noting that, even though SegGPT only requires a single segmentation sample for inference, it is a fully supervised method and demands a large segmentation dataset for training. In contrast, *SLiMe* is truly a *one-shot* technique, where only a single sample is needed for optimization.

**CelebAMask-HQ**. This is a dataset of the facial part segmentation, and we report results on the parts used in ReGAN for comparison (for more details, please consult Appendix B.1). Figure 6 and Table 4 showcase our qualitative and quantitative results. In the 1-sample setting, *SLiMe* outperforms other methods on average and for the majority of parts, demonstrating its superiority in 1-sample scenario. On the other hand, in the 10-sample setting, except for three parts, our method either performs better or comparably to other methods. As mentioned earlier, note that SegGPT benefits from training on

Table 4: **Segmentation results of CelebAMask-HQ10.** Our method consistently outperforms ReGAN, SegDDPM, and SegGPT in the majority of parts in 1-sample setting in the last four rows. Additionally, *SLiMe* either outperforms or performs comparably to ReGAN and SegDDPM in 10-sample setting in the first three rows. $^\star$ is used to denote supervised methods.

| | Cloth | Eyebrow | Ear | Eye | Hair | Mouth | Neck | Nose | Face | Background | Average |
|---|---|---|---|---|---|---|---|---|---|---|---|
| ReGAN | 15.5 | **68.2** | 37.3 | **75.4** | 84.0 | **86.5** | 80.3 | **84.6** | **90.0** | 84.7 | 69.9 |
| SegDDPM | 61.6 | 67.5 | **71.3** | 73.5 | **86.1** | 83.5 | 79.2 | 81.9 | 89.2 | 86.5 | **78.0** |
| *SLiMe* | **63.1 ± 1.6** | 62.0 ± 1.6 | 64.2 ± 1.9 | 65.5 ± 3.0 | 85.3 ± 0.4 | 82.1 ± 1.6 | 79.4 ± 2.2 | 79.1 ± 1.4 | 88.8 ± 0.2 | **87.1 ± 0.0** | 75.7 ± 0.4 |
| ReGAN | - | - | - | 57.8 | - | 71.1 | - | 76.0 | - | - | - |
| SegGPT$^\star$ | 24 | **48.8** | 32.3 | 51.7 | **82.7** | 66.7 | 77.3 | 73.6 | 85.7 | 28.0 | 57.1 |
| SegDDPM | 28.9 | 46.6 | **57.3** | **61.5** | 72.3 | 44.0 | 66.6 | 69.4 | 77.5 | 76.6 | 60.1 |
| *SLiMe* | **52.6 ± 1.4** | 44.2 ± 2.1 | 57.1 ± 3.6 | 61.3 ± 4.6 | 80.9 ± 0.5 | **74.8 ± 2.9** | **78.9 ± 1.3** | 77.5 ± 1.8 | **86.8 ± 0.3** | **81.6 ± 0.8** | **69.6 ± 0.3** |

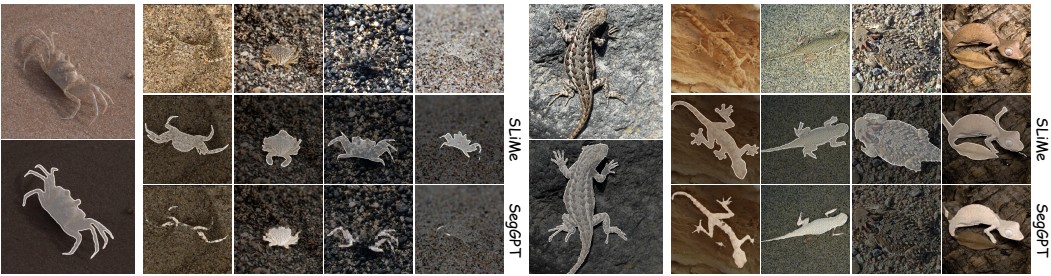

Figure 5: **Segmentation results of camouflaged objects.** The larger images are used for optimizing *SLiMe*, and as the source image for SegGPT. Notably, *SLiMe* outperforms SegGPT.

a large segmentation dataset. Also, the other two methods employ class-specific pre-trained models. In contrast, *SLiMe* utilizes a model pre-trained on general data, equipping it with the ability to work across a wide range of categories rather than being limited to a specific class.

**Additional Results**. We also showcase the versatility of our method, which can be optimized on an occluded object and infer images without the occlusion, or conversely, be optimized on a fully visible object and make predictions on occluded objects. This shows our method's capability to comprehend part and object semantics. Figure 11 illustrates that despite occlusion of the target region caused by the person in the image used for optimization, our method performs well. It is also possible to segment occluded objects using a visible reference object (see Figure 12). Moreover, in Figure 5, we compare our method against SegGPT (Wang et al., 2023) using two camouflaged animals, namely a crab and a lizard. Remarkably, *SLiMe* achieves precise segmentation of these animals, even in situations where they were challenging to be detected with naked eye. This shows that *SLiMe* learns rich semantic features about the target object that do not fail easily due to the lack of full perception.

## 6 CONCLUSION

We proposed *SLiMe*, a *one-shot* segmentation method capable of segmenting *various objects/parts* in *various granularity*. Through an extensive set of experiments and by comparing it to state-of-the-art few-shot and supervised image segmentation methods, we showed its superiority. We showed that, although *SLiMe* does not require training on a specific class of objects or a large segmentation dataset, it outperforms other methods. On the other hand, *SLiMe* has some limitations. For example, it may result in noisy segmentations when the target region is tiny. This can be attributed to the fact that the attention maps, which we extract from SD for segmentation mask generation, have a smaller size than the input image. To counter this, we employed bilinear interpolation for upscaling. Nonetheless, due to scaling, some pixels might be overlooked, leading to the undesired noisy outcomes. For visual examples of this case, please refer to Appendix A.1. Resolving the mentioned limitation, and making it applicable to 3D and videos, would be an interesting future direction.

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

# A    APPENDIX

## A.1    ADDITIONAL RESULTS

Figure 6 showcase our qualitative and quantitative results

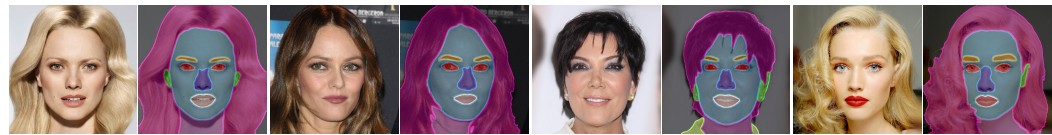

Figure 6: **Qualitative face segmentation results.** Results of *SLiMe* optimized with 10 samples.

In Figure 7, we provide comparisons with ReGAN (Tritrong et al., 2021). It is evident that *SLiMe* exhibits more intricate hair segmentation in the second and third rows, showcasing its ability to capture finer details compared to ReGAN. Additionally, in the second row, the ear segmentation produced by ReGAN appears to be noisy by comparison.

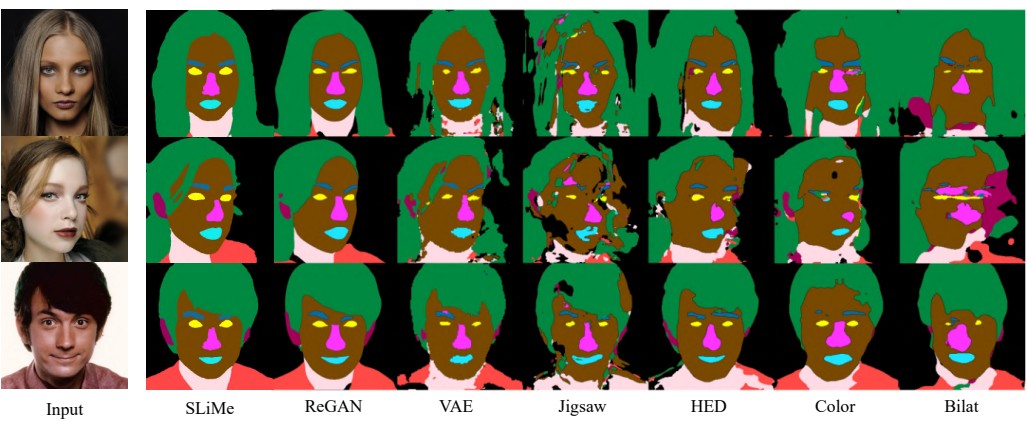

Input    SLiMe    ReGAN    VAE    Jigsaw    HED    Color    Bilat

Figure 7: **Qualitative comparison with other methods on CelebAHQ-Mask.** Qualitative results of several methods on the 10-sample setting of CelebAHQ-Mask. As you can see, *SLiMe* captures the details better than ReGAN and other methods (e.g., hairlines in the second row). All the images are taken from (Tritrong et al., 2021).

In addition to the segmentation results presented for several object categories in the paper, Figure 8 showcases additional 1-sample visual results. These results encompass a wide range of objects and provide evidence of *SLiMe*'s capability to perform effectively across various categories.

Another noteworthy feature of *SLiMe* is its generalization capacity, as illustrated in Figure 9. This figure demonstrates, despite being optimized on a single dog image with segmented parts, *SLiMe* can grasp the concepts of head, body, legs, and tail and effectively apply them to unseen images from various categories. However, Figure 10 illustrates that when optimized on an image containing both a dog and a cow, *SLiMe* is adept at learning to exclusively segment the dog class in unseen images. These two figures, highlight *SLiMe*'s ability to acquire either high-level concepts or exclusive object classes.

One more appealing feature of *SLiMe* is that not only is it able to learn to segment from an occluded image and segment fully visible objects (Figure 11), but it can also be optimized on a fully visible object and make predictions on occluded samples. As an example, in Figure 12, *SLiMe* is optimized on a fully visible bear and predicts an accurate segmentation mask for the occluded bears.

### A.1.1    FAILURE CASE OF *SLiMe*

As mentioned in the paper, *SLiMe* may fail in the cases where the target region is tiny. Figure 13 shows two examples of this, where the target to be segmented is a necklace, which is pretty small.

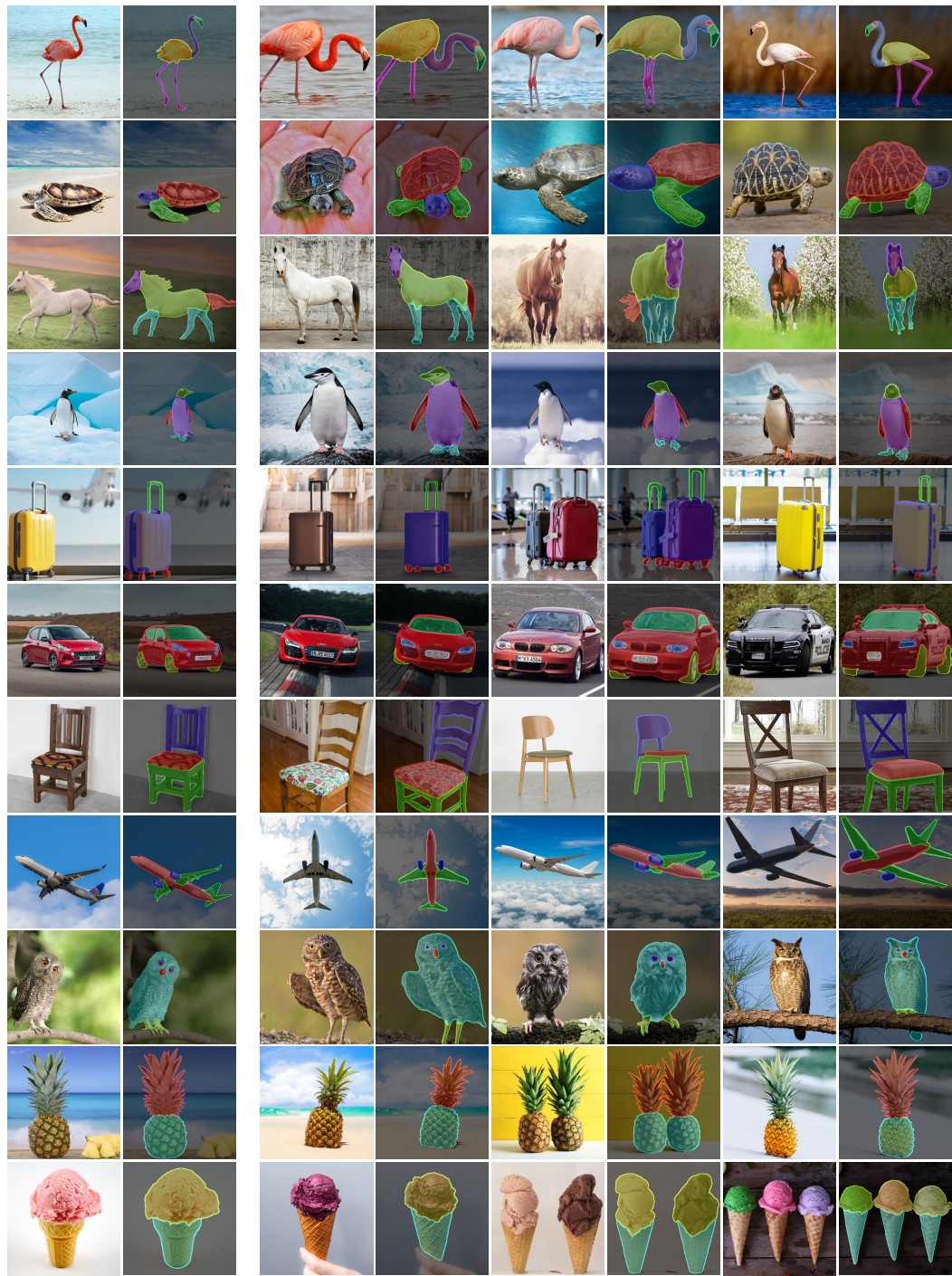

Figure 8: **Part segmentation results on different objects.** *SLiMe* exhibits strong performance across a wide variety of objects. The images, along with their corresponding annotations used for optimization, are displayed on the left.

## A.2 ABLATION STUDIES

In this section, we present the results of our ablation studies, which aim to illustrate the impact of each component in our model. These experiments were conducted using 10 samples from the car class in the PASCAL-Part dataset.

In Table 5, we present the results of text prompt ablation experiments to demonstrate the robustness of *SLiMe* to the choice of the initial text prompt. The "part names" row displays results obtained by using specific part names in the prompts. In the next row, labeled " ", the text prompt is left empty.

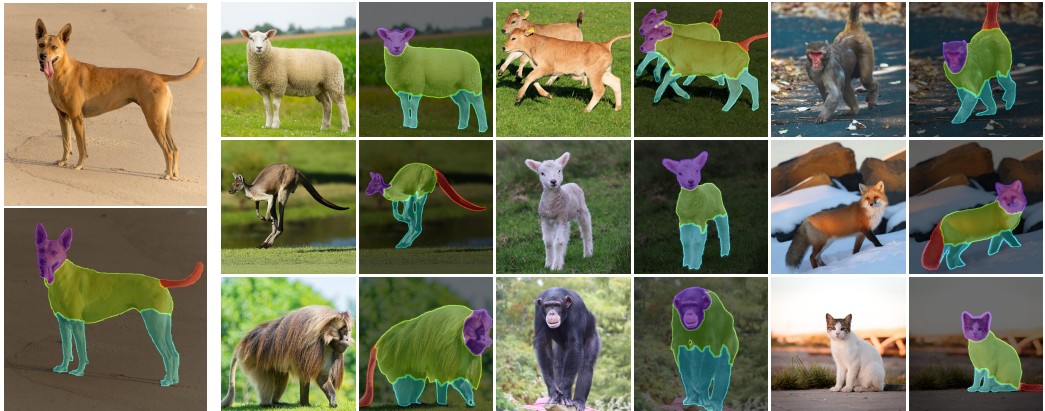

Figure 9: **Generalizability of *SLiMe*.** *SLiMe* optimized on dog's parts, can accurately segment corresponding parts of other animals.

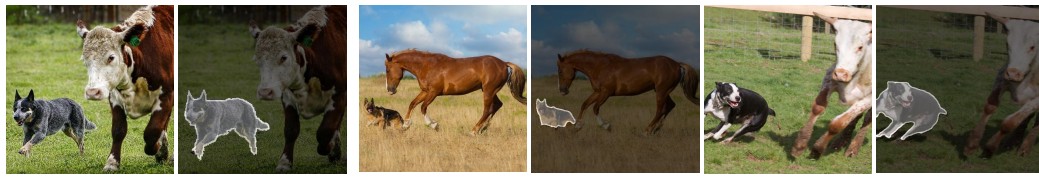

Figure 10: ***SLiMe* exclusive segmentation.** *SLiMe* optimized on an image containing both dog and cow (as seen in the left image pair), can segment only dogs, even in presence of other animals.

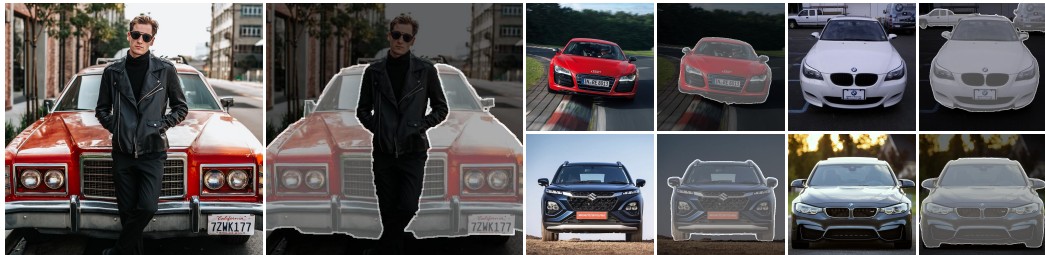

Figure 11: **Segmentation results of occluded objects.** Although *SLiMe* is optimized using an occluded car's image (the leftmost image), it demonstrates proficiency in car segmentation on unseen images (the remaining images on the right). Particularly noteworthy is its ability to accurately segment all three cars in the top-right image.

In the final row, we use the term "part" instead of specific part names. By comparing these rows, we observe minimal influence of the initial text prompt on the results.

Moving on, we conducted experiments to determine the optimal coefficients for our loss functions. As illustrated in the first four rows of Table 6, where $\alpha = 1$, the most suitable coefficient for $\mathcal{L}_{SD}$ is found to be $0.005$. Furthermore, when comparing the $3^{rd}$ and $5^{th}$ rows, the significance of $\mathcal{L}_{MSE}$ becomes apparent.

Next, we turn our attention to ablation of the parameters $t_{\text{trian}}$ and $t_{\text{test}}$. Initially, we vary the range used to sample $t_{\text{opt}}$. Table 7 shows that the best range is $[5, 100]$, which introduces a reasonable amount of noise. A larger end value in the range results in very noisy images in some steps, making it difficult to optimize the text embeddings. Conversely, a very small end value means that *SLiMe* does not encounter a sufficient range of noisy data for effective optimization.

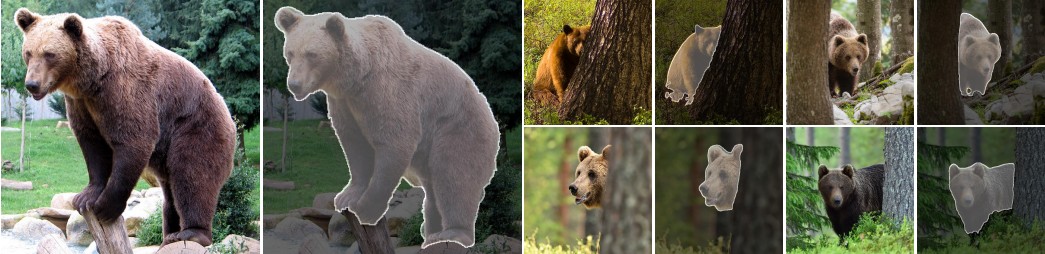

Figure 12: **Occluded object in inference.** *SLiMe* undergoes its initial optimization with a bear image, as depicted in the left image. Subsequently, it is put to the test with images featuring occluded portions of the bear. Notably, *SLiMe* precisely segments these occluded objects.

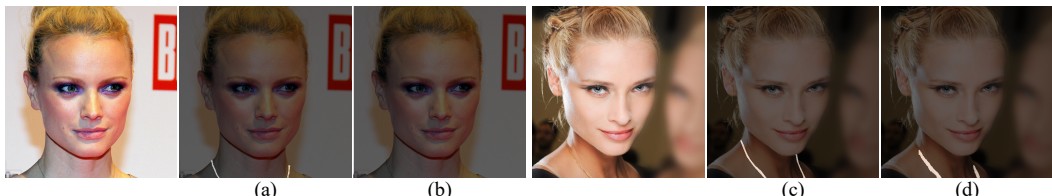

Figure 13: **Failure case.** The segmentation masks generated by *SLiMe*, depicted in images (b) and (d), reveal an inherent challenge. Our method encounters difficulty when it comes to accurately segmenting minuscule objects, such as the necklace in this image. These tiny objects often diminish in size, and at times, even vanish within the cross-attention maps we employ, primarily due to their limited resolution.

After examining $t_{\text{opt}}$, we ablate the parameter $t_{\text{test}}$. Selecting an appropriate value for $t_{\text{test}}$ is crucial, as demonstrated in Table 8. *SLiMe* performs optimally when we set $t_{\text{test}}$ to 100.

Another parameter subjected to ablation is the learning rate. Choosing the correct learning rate is essential, as a high value can result in significant deviations from text embeddings that SD can comprehend. Conversely, a low learning rate may not introduce sufficient changes to the embeddings. Our experiments in Table 9 reveal that the optimal learning rate for our method is $0.1$.

Finally, we performed an ablation study on the choice of layers to utilize their cross-attention modules. Based on our experiments in Table 10, we determined that the best set of layers to use are the $8^{th}$ to $12^{th}$ layers.

# B  IMPLEMENTATION DETAILS

We opted for SD version 2.1 and extracted the cross-attention and self-attention maps from the $8^{th}$ to $12^{th}$ and last three layers of the UNet, respectively. For the text prompt, we use a sentence where the word "prompt" is repeated by the number of parts to be segmented.

During optimization, we assigned a random value to the time step of SD's noise scheduler, denoted as $t_{\text{opt}}$, for each iteration. This value was selected randomly between 5 and 100, where $t_{\text{opt}}$ can be in a range spanning from 0 to 1000. During inference, we consistently set $t_{\text{test}}$ to 100.

For optimization, we employed the Adam optimizer with a learning rate of 0.1, optimizing our method for 200 epochs with a batch size of 1. Additionally, we used weighted cross-entropy loss, with each class's weight determined as the ratio of the number of whole pixels in the image to the number of pixels belonging to that class within the image. Furthermore, the values for $\alpha$ and $\beta$ were set to 1 and 0.005, respectively.

We set $H''$ and $W''$ to be 64. Regarding the CelebAMask-HQ, we divided the $512 \times 512$ images into 4 patches of size 400. After acquiring their WAS-attention maps from *SLiMe*, we aggregated them to generate the final WAS-attention map and subsequently derive the segmentation mask.

Table 5: **Ablating the text prompt.** Minimal effect of the initial text prompt for *SLiMe*. "*part names*": using all part names separated with space for text prompt. ("*background body light plate wheel window*"); " ": leaving the text prompt empty; "part": using "*part*" instead of part names ("*part part part part part part*"); *SLiMe* is with the second settings.

| Text Prompt | Body | Light | Plate | Wheel | Window | BG | Average |
|---|---|---|---|---|---|---|---|
| "*part names*" | **82.0 ± 0.3** | 55.3 ± 2.0 | **56.1 ± 1.1** | 69.4 ± 0.6 | 69.6 ± 0.9 | 79.5 ± 1.1 | **68.7 ± 0.4** |
| " " | 81.6 ± 1.0 | 56.7 ± 0.4 | 54.4 ± 3.5 | **69.6 ± 1.3** | 68.1 ± 0.6 | **80.2 ± 0.9** | 68.4 ± 1.2 |
| "*part*" | 81.5 ± 1.0 | **56.8 ± 1.2** | 54.8 ± 2.7 | 68.3 ± 0.1 | **70.3 ± 0.9** | 78.4 ± 1.6 | 68.3 ± 1.0 |

Table 6: **Ablating the loss terms.** Comparing the first four rows shows the importance of $\mathcal{L}_{SD}$. Furthermore, when comparing the last two rows, it underscores the effectiveness of $\mathcal{L}_{MSE}$.

| $\alpha$ | $\beta$ | Body | Light | Plate | Wheel | Window | BG | Average |
|---|---|---|---|---|---|---|---|---|
|  | 0.5 | 0.0 ± 0.0 | 0.0 ± 0.0 | 0.0 ± 0.0 | 0.0 ± 0.0 | 0.0 ± 0.0 | 31.8 ± 0.0 | 5.3 ± 0.0 |
| 1 | 0.05 | 80.1 ± 0.4 | **57.6 ± 0.1** | 46.3 ± 0.2 | 67.4 ± 1.0 | 63.0 ± 4.1 | 78.0 ± 0.2 | 65.4 ± 0.9 |
|  | 0.005 | **81.5 ± 1.0** | 56.8 ± 1.2 | 54.8 ± 2.7 | 68.3 ± 0.1 | **70.3 ± 0.9** | **78.4 ± 1.6** | **68.3 ± 1.0** |
|  | 0.0 | 73.7 ± 2.3 | 39.7 ± 1.2 | 38.0 ± 3.1 | 55.0 ± 3.4 | 61.3 ± 4.0 | 67.2 ± 2.2 | 55.8 ± 1.6 |
| 0 | 0.005 | 80.6 ± 0.5 | 56.1 ± 1.1 | **57.4 ± 0.4** | **69.1 ± 0.4** | 66.7 ± 1.6 | 78.0 ± 1.4 | 68.0 ± 0.3 |

When constructing the WAS-attention map from the cross-attention and self-attention maps, we only considered the corresponding cross-attention map of a token if the maximum value in that map exceeded 0.2. Otherwise, we disregarded that token and assigned zeros to its corresponding channel in the WAS-attention map.

For optimizing on PASCAL-Part classes, we applied the following augmentations: Random Horizontal Flip, Gaussian Blur, Random Crop, and Random Rotation. For the car class, we set the random crop ratio range to $[0.5, 1]$, while for the horse class, it was adjusted to $[0.8, 1]$. Additionally, we applied random rotation within the range of $[-30, 30]$ degrees.

Moreover, when optimizing on CelebAMask-HQ, we incorporated a set of augmentations, which encompassed Random Horizontal Flip, Gaussian Blur, Random Crop, and Random Rotation. The random crop ratio was modified to fall within the range of $[0.6, 1]$, and random rotation was applied within the range of $[-10, 10]$ degrees.

## B.1 DETAILS OF THE DATASETS

In this section, we initially present further elaboration on the datasets on which we optimized our method. This includes information about the categories as well as the segmentation labels. Subsequently, we offer details about the dataset preparation process.

### B.1.1 PASCAL-PART

- **Car**: Background, Body, Light, Plate, Wheel, and Window.
- **Horse**: Background, Head, Leg, Neck+Torso, and Tail.

### B.1.2 CELEBAMASK-HQ

- **Face**: Background, Cloth, Ear, Eye, Eyebrow, Face, Hair, Mouth, Neck, and Nose.

### B.1.3 DATASET PREPARATION

- **PASCAL-Part**. For this dataset, we follow the procedures of ReGAN (Tritrong et al., 2021): We start by cropping the images with the bounding boxes provided in the dataset. Afterward, we remove those images where their bounding boxes have an overlap of more

Table 7: **Ablating $t_{\text{trian}}$.** Our results across different ranges for choosing $t_{\text{opt}}$ indicate that optimal performance is achieved when $t_{\text{opt}}$ is selected from the range $[5, 100]$.

| Range of $t_{\text{opt}}$ | Body | Light | Plate | Wheel | Window | BG | Average |
|---|---|---|---|---|---|---|---|
| $[5, 20]$ | $78.8 \pm 1.6$ | $52.6 \pm 2.8$ | $53.6 \pm 1.0$ | $66.6 \pm 0.2$ | $\mathbf{70.3 \pm 0.1}$ | $78.7 \pm 1.2$ | $66.8 \pm 0.2$ |
| $[5, 100]$ | $\mathbf{81.5 \pm 1.0}$ | $\mathbf{56.8 \pm 1.2}$ | $\mathbf{54.8 \pm 2.7}$ | $\mathbf{68.3 \pm 0.1}$ | $70.3 \pm 0.9$ | $78.4 \pm 1.6$ | $\mathbf{68.3 \pm 1.0}$ |
| $[5, 900]$ | $80.4 \pm 1.1$ | $54.4 \pm 1.6$ | $54.3 \pm 2.4$ | $67.3 \pm 1.3$ | $70.2 \pm 0.4$ | $\mathbf{79.9 \pm 1.7}$ | $67.8 \pm 1.0$ |

Table 8: **Ablating $t_{\text{test}}$.** Evident from the table, we get the best results when $t_{\text{test}} = 100$

| $t_{\text{test}}$ | Body | Light | Plate | Wheel | Window | BG | Average |
|---|---|---|---|---|---|---|---|
| 5 | $80.3 \pm 0.4$ | $50.8 \pm 1.8$ | $49.7 \pm 4.2$ | $66.5 \pm 0.8$ | $65.5 \pm 0.7$ | $\mathbf{78.4 \pm 1.2}$ | $65.2 \pm 1.4$ |
| 20 | $80.1 \pm 0.8$ | $53.1 \pm 2.7$ | $53.6 \pm 1.9$ | $65.6 \pm 2.7$ | $67.3 \pm 2.1$ | $76.0 \pm 2.5$ | $65.9 \pm 0.7$ |
| 100 | $\mathbf{81.5 \pm 1.0}$ | $\mathbf{56.8 \pm 1.2}$ | $\mathbf{54.8 \pm 2.7}$ | $\mathbf{68.3 \pm 0.1}$ | $\mathbf{70.3 \pm 0.9}$ | $78.4 \pm 1.6$ | $\mathbf{68.3 \pm 1.0}$ |

than $5\%$ with other bounding boxes. Finally, we remove the cropped images smaller than $50 \times 50$ for the car class and $32 \times 32$ for the horse class.

- **CelebAMask-HQ**. The size of images that we use for this dataset is $512 \times 512$.

Table 9: **Ablating lr.** The results of optimizing *SLiMe* with various learning rates reveal a crucial relationship. When the learning rate is set too low, *SLiMe* struggles to learn effectively, resulting in minimal progress. Conversely, when the learning rate is excessively high, the text embeddings deviate significantly from the comprehensible embeddings of SD.

| lr | Body | Light | Plate | Wheel | Window | BG | Average |
|----|------|-------|-------|-------|--------|-----|---------|
| 1 | $10.1 \pm 9.2$ | $3.6 \pm 6.2$ | $12.4 \pm 10.8$ | $11.2 \pm 11.2$ | $13.2 \pm 14.5$ | $39.7 \pm 11.4$ | $15 \pm 7.4$ |
| 0.1 | $81.5 \pm 1.0$ | $\mathbf{56.8 \pm 1.2}$ | $\mathbf{54.8 \pm 2.7}$ | $68.3 \pm 0.1$ | $\mathbf{70.3 \pm 0.9}$ | $78.4 \pm 1.6$ | $\mathbf{68.3 \pm 1.0}$ |
| 0.01 | $\mathbf{81.5 \pm 0.0}$ | $54.9 \pm 0.4$ | $52.8 \pm 1.7$ | $\mathbf{68.5 \pm 0.5}$ | $69.8 \pm 0.2$ | $\mathbf{79.2 \pm 0.6}$ | $67.8 \pm 0.3$ |
| 0.001 | $70.0 \pm 0.9$ | $10.5 \pm 5.1$ | $40.6 \pm 0.5$ | $0.7 \pm 0.5$ | $18.2 \pm 6.1$ | $62.4 \pm 1.6$ | $33.7 \pm 2.3$ |

Table 10: **Ablating layers to use their cross-attention module.** The middle layers of SD's UNet exhibit a superior semantic understanding compared to the other set of layers.

| set of cross attention layers | Body | Light | Plate | Wheel | Window | BG | Average |
|------------------------------|------|-------|-------|-------|--------|-----|---------|
| $1^{st}$ to $7^{th}$ layers | $12.7 \pm 9.5$ | $13.8 \pm 12.1$ | $10.7 \pm 3.7$ | $29.5 \pm 14.2$ | $32.1 \pm 2.9$ | $34.2 \pm 4.1$ | $22.2 \pm 3.4$ |
| $8^{th}$ to $12^{th}$ layers | $\mathbf{81.5 \pm 1.0}$ | $\mathbf{56.8 \pm 1.2}$ | $54.8 \pm 2.7$ | $\mathbf{68.3 \pm 0.1}$ | $\mathbf{70.3 \pm 0.9}$ | $\mathbf{78.4 \pm 1.6}$ | $\mathbf{68.3 \pm 1.0}$ |
| $13^{th}$ to $16^{th}$ layers | $76.8 \pm 0.8$ | $51.9 \pm 9.2$ | $\mathbf{56.2 \pm 3.0}$ | $62.6 \pm 4.2$ | $65.3 \pm 2.7$ | $68.6 \pm 2.6$ | $63.6 \pm 1.4$ |

