# OpenReview forum: "SLiMe: Segment Like Me"
_ICLR.cc/2024/Conference — ICLR 2024 poster_

### Official Review · Reviewer_16rP · 2023-10-31

**Soundness:** 3 good
**Presentation:** 3 good
**Contribution:** 3 good
**Rating:** 8
**Confidence:** 3

**Summary:**

This paper examines the problem of class-specific semantic part segmentation, under a one/few-shot data setting.

The method proposed by the paper is to leverage recent advances and findings in diffusion models, specifically in stable diffusion the cross-attention modules capturing relevant spatial regions and being used for semantic correspondence.

Learning is done by fine-tuning the text embedding from a stable diffusion model on an image and a corresponding part segmentation mask.  The embedding is optimized under a loss with three components, encouraging the cross-attention map, and introduced weighted accumulated self-attention map, to match the ground-truth segmentation, while not straying too far from the original stable diffusion loss.

Experimental validation is done on car, horse, and face part segmentation, with comparable to favorable results when compared against several recent state-of-the-art baselines.

**Strengths:**

- a nice framework for tackling the problem of one/few-shot semantic part segmentation, having clear benefits over existing proposed approaches in terms of amount of additional supervised annotations required for training, while maintaining similar performance

- in general, paper details are clearly presented, including a thorough supplementary appendix

**Weaknesses:**

- one of the stated contributions is the introduced weighted accumulated self-attention map.  This seems important enough that the incremental contribution of this component to the overall performance should perhaps be added to the main text rather than deferred to the appendix.

- further, within the appendix, Table 9 shows an improvement from adding WAS-attention, from 62.7 on average to 68.3.  I'm a little confused, then, on how this differs from Table 5, as I would have though the last row, setting $\alpha=0$, would also correspond to dropping WAS-attention, and here the average performance is 68.0

- lastly, on initial read, I was unsure of how text/text prompt was being used within the proposed method.  This was clarified by the first ablation study in A.2, but perhaps a sentence mention of this in the main text would also be helpful.

**Questions:**

see weaknesses above, in particular the issue raised in the second bullet

---

> ### Author Response · Authors · 2023-11-18
>
> **Clarification on Tables 5 and 9**
>
> We apologize for the confusion. We clarify that WAS-attention can be employed either solely during inference or during both optimization and inference. When used solely during inference, it has no impact on the optimization process.
>
> The reference made by the reviewer to Table 5 refers to the scenario where WAS-attention is applied exclusively during inference. On the other hand, Table 9 presents results when WAS-attention is entirely excluded from both optimization and inference steps in the car parts segmentation task. Notably, this exclusion leads to a notable decrease in IoU, dropping from 68.3% to 62.7%.
>
> We ran another experiment with completely removing WAS-attention from both optimization and inference steps on the horse parts segmentation task, where we also see a significant IoU drop (63.3% to 52.4%).
>
> This table provides a clear breakdown of the values for each body part, comparing the results without and with the Weighted Average Score (WAS).
>
> | Part      | Without WAS   | With WAS      |
> |----------------|---------------|---------------|
> | Background     | 68.2% ± 5.0   | 79.6% ± 2.5   |
> | Head           | 47.8% ± 2.8   | 63.8% ± 0.7   |
> | Legs           | 50.1% ± 0.6   | 59.5% ± 2.1   |
> | Neck+Torso     | 57.3% ± 13.1  | 68.1% ± 4.4   |
> | Tail           | 38.6% ± 1.4   | 45.4% ± 2.4   |
> | Average        | 52.4% ± 3.8   | 63.3% ± 2.4   |
>
> ==================
>
> **Clarification on prompt/tokens**
>
> Please refer to our answer to Reviewer kAQy

---

> > ### Comment · Reviewer_16rP · 2023-11-21
> >
> > Thank you for the helpful replies, both to myself and the other reviewers, this clarifies my confusion and addresses my concerns.

---

### Official Review · Reviewer_VRf7 · 2023-11-05

**Soundness:** 3 good
**Presentation:** 3 good
**Contribution:** 3 good
**Rating:** 8
**Confidence:** 4

**Summary:**

This work proposes a one-shot segmentation approach using the stable diffusion model. They pose the problem as one-shot optimization to perform object segmentation at different granularity levels conditioned on a single segmentation map. They take advantage of the self / cross-attention layer in the diffusion model to optimize the text embeddings for the given semantic segmentation task. They evaluate the proposed method on two public datasets and show its outperformance against the recent related work.

**Strengths:**

1. The proposed idea of optimizing the text embedding based on the attention maps for semantic segmentation is interesting and novel.

2. The proposed method is shown to be advantageous in two datasets both quantitatively and qualitatively.

3. The paper is well-written and easy to follow. The theoretical background is explained well.

4. The limitations are discussed.

5. The method is well-ablated for the different components.

**Weaknesses:**

1. The proposed method seems to be adapted for the segmentation task based on the work by Hedlin et al for unsupervised semantic correspondence.

2. Related works which are not cited:
[a] Burgert, Ryan, et al. "Peekaboo: Text to image diffusion models are zero-shot segmentors." arXiv preprint arXiv:2211.13224 (2022).
[b] Tian, Junjiao, et al. "Diffuse, Attend, and Segment: Unsupervised Zero-Shot Segmentation using Stable Diffusion." arXiv preprint arXiv:2308.12469 (2023).

3. The number of works that are compared against is limited.

Minor:
1. Making the best result in the ablation study tables bold would improve the readability.

**Questions:**

1. Is there any specific reason behind not including the SegDDPM results in Tab. 1?

**Details Of Ethics Concerns:**

The paper is available on arXiv in ICLR template (mentioning "Under review as a conference paper at ICLR 2024"). I am not sure whether it is acceptable or not: https://arxiv.org/pdf/2309.03179.pdf

---

> ### Author Response · Authors · 2023-11-18
>
> **Hedlin et al.’s work**
>
> We acknowledge that the high level idea of our work might have similarities to Hedlin et al.’s work, but their objectives are completely different from ours: we perform image segmentation while their task is image correspondence. Furthermore, in terms of architectural similarities, the optimization of cross-attention maps has been previously explored for various downstream tasks, as demonstrated in the work by Hedlin et al. However, our current task differs, and adopting their approach for our segmentation task did not yield effective results. Hedlin et al.’s exploration is confined to the optimization of a single text embedding, while our approach allows optimizing multiple text embeddings, enabling the simultaneous segmentation of various parts or objects.
>
> We attribute the significantly enhanced performance of our method to the utilization of WAS-attention which we propose in this work, as demonstrated in Table 9 and also the following results on horse part segmentation which we ran to address reviewer Fjuo’s comment.
>
> | Part      | Without WAS   | With WAS      |
> |----------------|---------------|---------------|
> | Background     | 68.2% ± 5.0   | 79.6% ± 2.5   |
> | Head           | 47.8% ± 2.8   | 63.8% ± 0.7   |
> | Legs           | 50.1% ± 0.6   | 59.5% ± 2.1   |
> | Neck+Torso     | 57.3% ± 13.1  | 68.1% ± 4.4   |
> | Tail           | 38.6% ± 1.4   | 45.4% ± 2.4   |
> | Average        | 52.4% ± 3.8   | 63.3% ± 2.4   |
>
> ===================
>
> **Related works [a] and [b]**
>
> Thank you for bringing this to our attention. We've noticed that neither of the two works are peer-reviewed. While we appreciate their contributions, it's essential to highlight the distinctions that make them not directly comparable to SLiMe. In summary, Peekaboo requires an object description for each segment, a requirement absent in SLiMe. Additionally, Peekaboo lacks the capability to perform segmentation at the object part level, which is a fundamental feature of SLiMe.
>
> Regarding the second work by Junjiao et al., as we understand it, the method primarily segments images into different parts based on feature similarity, akin to the approach in the Segment Anything paper. This method lacks control over segmentation granularity and cannot perform targeted object/part segmentation. For example, it cannot be directed to segment a specific part of an object. These distinctions emphasize the unique capabilities and advantages of SLiMe in comparison.
>
>
> =================
>
> **The number of methods that are compared against are limited**
>
> We have conducted thorough comparisons with three _most_ relevant papers to SLiMe. Notably, these papers also provide comparisons with other existing methods, offering readers valuable insights into our performance relative to those cited works (although we didn't include them in our study as their setups differed from ours). In response to a reviewer's suggestion (please refer to our responses to reviewer Fjuo for details), we further conducted experiments on two additional datasets, providing readers with a broader context for evaluating the performance of our work. We are open to suggestions on specific papers that you believe warrant comparison, and we genuinely welcome any recommendations for meaningful comparisons to be incorporated into the final version of the paper. Thank you
>
> =================
>
> **Not including SegDDPM in Table 1**
>
> We conducted a comparative analysis of SLiMe and SegDDPM on two segmentation tasks: _face_ and _horse_ part segmentation (Tables 2 and 3). It's worth noting that the SegDDPM paper did not report results for the _car_ part segmentation task (Table 1). Our attempts to run their method on this task revealed a challenge: for SegDDPM to segment cars into their parts, it requires a DDPM trained to generate car images. Unfortunately, the SegDDPM authors have not provided a DDPM checkpoint for car image generation, preventing us from assessing SegDDPM's performance in car part segmentation.

---

> > ### Comment · Reviewer_VRf7 · 2023-11-23
> >
> > Thank you for your response. It clarified most of my questions and concerns.

---

### Official Review · Reviewer_Fjuo · 2023-11-06

**Soundness:** 3 good
**Presentation:** 3 good
**Contribution:** 3 good
**Rating:** 6
**Confidence:** 2

**Summary:**

This paper proposes to retarget the Stable Diffusion model (SD) for few-shot semantic segmentation. Instead of taking the in-context learning or data generation approaches, this paper proposes a new pipeline which optimizes the text embedding in SD on the input image to "find" the text embedding to correspond to the segmented region. In addition to the cross-attention map of the optimized text token, it proposes a new self-attention map fusion module to regress the ground truth mask with a higher resolution. The proposed method achieves SOTA result on the benchmarks used in one previous work. The ablation studies shows the effectiveness of the model design.

**Strengths:**

- A novel and interesting idea. The idea of retargeting the feature representation in a pretrained generative model for few-shot semantic segmentation is not new. But it's novel to exploit the text branch in the text-based image generation model (i.e. Stable Diffusion). Instead of training the adaptor model to map between the generative features to the semantic masks which may overfit on input image features, the proposed method aims to find the text embedding which may be more generalizable.

- The proposed method significantly outperforms SOTA result on the benchmarks used in the ReGAN paper. The ablation studies shows the effectiveness of the model design.

- The paper provides enough details in the appendix for reproduction and the open-sourced code is straightforward to follow.

**Weaknesses:**

- Unconvincing importance of WAS attention: Figure 1 shows a good intuition that we need WAS attention to refine the object boundary. However, In Table 5, the contribution of the WAS attention doesn't seem to be significant. For the without WAS attention results (fig. 2a, 2c), will they be much improved by using GrabCut or other methods for segmentation refinement post-processing?

- Lack of comparison on benchmark datasets like ADE-Bedroom-30 (used by segDDPM) and FSS-1000 (used in segGPT). The current quantitative results are only on horse/car/face datasets used in ReGAN. The result can be more solid with evaluation on diverse types of objects/parts.

- Figure 3 is a confusing. Currently, it seems like the predicted noise is from the cross-attention map and the WAS-attention map. It'll be better to put the Unet from SD in the box. Then from this Unet, one output is the predicted noise from the original SD and the other output is fed to the attention-extraction module which selects layers and combines attention maps.

**Questions:**

- For the learned text embedding, are they interpretable? e.g. one can use the data-driven approach to find text tokens whose embedding are closest to the optimized ones.

- How is the performance on other benchmark datasets like ADE-Bedroom-30 (used by segDDPM) and FSS-1000 (used in segGPT)

---

> ### Author Response · Authors · 2023-11-18
>
> **WAS-attention's effect**
>
> We would like to clarify that WAS-attention can be employed either solely during inference or during both optimization and inference. When used solely during inference, it has no impact on the optimization process.
>
> The reference made by the reviewer to Table 5 refers to the scenario where WAS-attention is applied exclusively during inference. On the other hand, Table 9 presents results when WAS-attention is entirely excluded from both optimization and inference steps in the car parts segmentation task. Notably, this exclusion leads to a notable decrease in IoU, dropping from 68.3% to 62.7%.
>
> We ran another experiment with completely removing WAS-attention from both optimization and inference steps on the horse parts segmentation task, where we also see a significant IoU drop (63.3% to 52.4%).
>
> This table provides a clear breakdown of the values for each body part, comparing the results without and with the Weighted Average Score (WAS).
>
> | Part      | Without WAS   | With WAS      |
> |----------------|---------------|---------------|
> | Background     | 68.2% ± 5.0   | 79.6% ± 2.5   |
> | Head           | 47.8% ± 2.8   | 63.8% ± 0.7   |
> | Legs           | 50.1% ± 0.6   | 59.5% ± 2.1   |
> | Neck+Torso     | 57.3% ± 13.1  | 68.1% ± 4.4   |
> | Tail           | 38.6% ± 1.4   | 45.4% ± 2.4   |
> | Average        | 52.4% ± 3.8   | 63.3% ± 2.4   |
>
> =====================
>
> **Post-processing e.g., GrabCut**
>
> When it comes to post-processing techniques like traditional GrabCut, it's important to note that not only do they introduce additional computations during inference, but they may also necessitate user intervention for refining segmentations in complex cases (refer to Figure 5 in the GrabCut paper). These methods often struggle to generalize in scenarios where there is no distinct color difference between object and background, or in cases with subtle color/texture transitions between different parts of an object. For instance, in face part segmentation, distinguishing noise from the actual face becomes challenging due to their almost identical texture and color.
>
> =====================
>
> **Results on ADE-Bedroom-30 and FSS-1000**
>
> We would like to emphasize that while the SegGPT model was tested on unseen samples/datasets, its foundational base model is trained on extensive supervised segmentation data—comprising images and corresponding segmentation masks. This means that SegGPT's underlying supervised model may have encountered categories or samples resembling the test data. However, it's important to note that we do not utilize a base supervised segmentation model. Nevertheless, when applied to the FSS-1000 dataset, our method yielded a reasonable IoU:
>
> SLiMe (ours): 73.53 %
> Painter: 61.7%
> SegGPT: 85.6%
>
> Painter is cited in the SegGPT paper as a similar technique which also leverages a supervised segmentation base model (as SegGPT does). Notably, our SLiMe outperforms Painter.
>
> For the ADE-Bedroom-30 dataset, we obtained an average mIoU of 32.3 using the same settings as SegDDPM, which is comparable to the 34.6 achieved by segDDPM. We identified a slight underperformance in SLiMe, attributed to its imperfect handling of small and featureless objects, such as man-made items like lamps in the ADE-Bedroom dataset. This limitation arises from the fact that these objects become tiny or even vanish in the cross-attention maps we employ, sized at 16x16 and 32x32, while the input images for this dataset are at a resolution of 256x256. Addressing this issue could be a valuable improvement for SLiMe in future work.
>
> =====================
>
> **Interpretability of learned embeddings**
>
> We calculated cosine distance between some of the learned text embeddings with our SLiMe and the words/embeddings in the Stable Diffusion dictionary. The closest text tokens in the Stable Diffusion’s dictionary to the learned ones using SLiMe do not necessarily match. For instance, the 10 closest words in the Stable Diffusion dictionary to the optimized embedding of 'bear' segmentation are:
>
> ```
> {brasil, nightout, mccall, boudo, confirmation, oldschool, stockholm, regardless, beacon, tik}
> ```
>
> which do not correspond to the concept of 'bear.' However, a future work could involve a deeper exploration of the initial and the evolution of the word embeddings during optimization, which may lead to improved segmentation results or more interpretable tokens.

---

### Official Review · Reviewer_kAQy · 2023-11-10

**Soundness:** 3 good
**Presentation:** 3 good
**Contribution:** 3 good
**Rating:** 6
**Confidence:** 3

**Summary:**

This paper presents SLiMe that allows for the segmentation of various objects or parts at different granularity levels with just one annotated example. The method leverages the knowledge embedded in pre-trained vision-language models and uses weighted accumulated self-attention maps and cross-attention maps to optimize text embeddings. The optimized embeddings then assist in segmenting unseen images, demonstrating the method's effectiveness even with minimal annotated data. The paper includes extensive experiments showing that SLiMe outperforms existing one- and few-shot segmentation methods.

**Strengths:**

1: SLiMe introduces a unique one-shot optimization strategy for image segmentation, which is useful when the available data is limited.

2: The proposed method demonstrates superior performance over existing one- and few-shot segmentation methods in various tests, indicating its practical applicability and effectiveness.

3: The paper showcases the method's versatility by successfully applying it to different objects and granularity levels, emphasizing its broad applicability.

**Weaknesses:**

My main concerns focus on the **text prompt**.

1: The introduction of the text prompt is quite abrupt. In the Introduction, SLiMe is described as requiring only an image and a corresponding mask to achieve segmentation of any granularity. However, immediately after, the author talks about fine-tuning text embeddings. What is the definition of 'text' in this task? How are text embeddings obtained? And do different granularities correspond to the same text? The author is encouraged to provide further clarification.

2: The role of "text prompt" in the method. The authors claim that "our novel WAS-attention map to fine-tune the text embeddings, enabling each text embedding to grasp semantic information from individual segmented regions". However, I haven't found evidence from the main text to illustrate the correspondence between "text embedding" and "individual segmented regions", especially in the arbitary granularity situation, which is one of the most important parts of this paper. Moreover, how to construct the text is also ignored.

**Questions:**

Please see the weakness part.

---

> ### Author Response · Authors · 2023-11-18
>
> **Clarification on prompt, tokens, and embedding**
>
> We appreciate the reviewer's insight regarding the potentially confusing tokenization step. To address this concern, we plan to include a new figure similar to this: https://imgur.com/a/0dutj8r that provides a clearer illustration of how each part relates to tokens for granular level segmentation. As illustrated in the figure, each part (p) in the image corresponds to a token (t) in the embedding, which is initially set as a null text (i.e., "") to be later optimized. Consequently, we do not utilize any prompt as input. The inputs to SLiMe include empty token placeholder(s), an image, and a corresponding segmentation mask.
>
> We utilize Stable Diffusion's text tokenizer to specify text tokens (i.e., breaking the sentence into tokens). Since Stable Diffusion is configured to operate with 77 tokens, if the length of the tokenized text is less than 77, the tokenizer pads it with a special token (e.g., end of string: EOS). If its length exceeds 77, the tokenizer crops it to a length of 77. We initialize the tokens with null text (i.e., ""). So they are like placeholders in the embedding.
>
> We hope this clarification resolves any confusion. Please feel free to let us know if any ambiguity persists.

---

> > ### Comment · Reviewer_kAQy · 2023-11-19
> >
> > Thanks for the detailed reply! The illustration in https://imgur.com/a/0dutj8r makes sense. For a better understanding, could authors provide some examples of how the learned text embeddings correspond to different parts of the real images? I think these samples could strengthen the claim.

---

> > > ### Author Response · Authors · 2023-11-19
> > >
> > > Thank you for your response. We present another example using a real sample [here](https://imgur.com/a/TDzUT2c), illustrating how each part is initially initialized as null text. Following optimization, each of these null texts transforms into an embedding that corresponds to a specific part in the single segmentation mask we provide as input.
> > >
> > > If the question relates to whether the embeddings for each part represent sensible or semantically meaningful words—the short answer is not necessarily. We conducted an experiment and identified the closest words in the Stable Diffusion space to the learned embedding. The learned embeddings do not match the closest words.  Please see our response to reviewer Fjuo under *Interpretability of learned embeddings*.

---

### Author Response · Authors · 2023-11-18
**General Response**

We appreciate the valuable and positive feedback from the reviewers. We will diligently address each reviewer's comments separately, and in the final version, we'll ensure to incorporate all editorial suggestions and figure fixes. Thank you again for your insightful input.

---

### Meta-Review · Area_Chair_F3by · 2023-12-11

**Metareview:**

The paper introduces a novel framework for one-shot or few-shot semantic segmentation, leveraging recent advances in diffusion models. The proposed method focuses on optimizing text embeddings through fine-tuning from a stable diffusion model using an image and its corresponding part segmentation mask. The paper excels in originality, consistently outperforming state-of-the-art methods across diverse benchmarks. Its universally praised clarity of presentation, versatility in application, and commitment to open science, including detailed information for reproducibility, are notable strengths. Discussions on limitations add credibility, and the technical soundness of methodologies, incorporating innovations like weighted accumulated self-attention maps, is recognized as a key strength.
Weaknesses include a lack of clarity on the concept of "text" and the role of text embeddings in the proposed methodologies, prompting reviewers to call for clearer definitions. Inconsistencies in reported results, particularly in the contribution of introduced components, raise concerns about the coherence of findings. Limited benchmark comparisons are a shared concern, with reviewers advocating for a more comprehensive evaluation on diverse datasets to enhance generalizability.

**Justification For Why Not Higher Score:**

While the results show promise and are solid, the idea of optimizing text embeddings through fine-tuning from a stable diffusion model using an image and its corresponding part segmentation mask may not be considered highly groundbreaking.

**Justification For Why Not Lower Score:**

The paper is distinguished by its originality, consistently surpassing state-of-the-art methods across diverse benchmarks. It has received widespread acclaim for its clear and accessible presentation, versatility in application, and commitment to open science.

---

### Decision · Program_Chairs · 2024-01-16

Accept (poster)